# Heidelberg Adult and Pediatric Airway Registry (HAPA-Registry)

**DOI:** 10.3390/mps8010006

**Published:** 2025-01-07

**Authors:** Davut D. Uzun, Kim Bittlinger, Emily Wibbe, Stefan Mohr, Markus A. Weigand, Felix C. F. Schmitt

**Affiliations:** Medical Faculty Heidelberg, Department of Anesthesiology, Heidelberg University, 69120 Heidelberg, Germany; kim.bittlinger@stud.uni-heidelberg.de (K.B.); emily.wibbe@stud.uni-heidelberg.de (E.W.); stefan.mohr@med.uni-heidelberg.de (S.M.); felix.schmitt@med.uni-heidelberg.de (F.C.F.S.)

**Keywords:** airway register, airway management, airway devices, tracheal intubation

## Abstract

Background: Advanced airway management is of fundamental importance in almost all areas of anesthesiology, emergency medicine, and critical care. Securing the airway is of the utmost importance, as this is a prerequisite for the oxygenation of the human organism. The clinical relevance of airway management is particularly evident in the fact that the primary cause of significant anesthesia-related complications can be attributed to this field. The need for the systematic recording of these procedures and their complications, as well as structured training in airway management and the evaluation of outcome parameters, is, therefore, evident. Methods: The HAPA-registry is a prospective and monocentric observational trial at the Department of Anesthesiology, Medical Faculty Heidelberg, University of Heidelberg, Germany. All patients requiring general anesthesia with consecutive advanced airway management during a surgical procedure were included. We, therefore, planned to include approximately 9000 patients in the first period. Following successful airway management, the anesthetist completed a case report form (CRF) in person. The intention was to record airway management cases on an annual basis for a period of several months, thus ensuring that the register remains up-to-date and that airway management procedures are continuously recorded. Discussion: The aim of this study was to establish an airway registry that enables the systematic recording and evaluation of different methods of airway management. The registry can be used to monitor and evaluate the implementation of current guidelines and recommendations for action, as well as the rates of success and complications.

## 1. Introduction

Ensuring a secure airway is a critical skill for anesthetists, as well as intensive care and emergency physicians, because the oxygenation of the human body relies on maintaining an open or secured airway. Advances in techniques and the adoption of guidelines and strategies for managing difficult airways significantly reduced morbidity and mortality rates [1,2]. The term “difficult airway” encompasses challenges encountered during advanced airway management. Although innovative methods like video laryngoscopy (VL) and fiberoptic intubation have been introduced, the definition of a “difficult airway” continues to be based on traditional tracheal intubation techniques using direct laryngoscopy (DL) [3]. Recent studies in airway management indicated that the primary use of VL in adult tracheal intubation is linked to higher success rates and fewer complications, such as hypoxemia [4,5,6]. The placement of an extraglottic airway (EGA) is considered difficult if several placement attempts are necessary [7]. Challenges with tracheal intubation are often broadly categorized as “difficult intubation”, without differentiating between difficulties in laryngoscopy and tracheal intubation. However, when indirect laryngoscopy methods like video laryngoscopy (VL) are employed, it becomes essential to distinguish between these two steps, as the occurrence of difficult laryngoscopy is consistently lower than that of difficult or failed tracheal intubation. The reported incidence of difficult direct laryngoscopy (DL) ranges from 1.5% to 8.0%, whereas the incidence of difficult intubation is lower [8]. Airway and respiratory complications account for approximately one-third of serious complications during anesthesia [9]. One of the most feared complications in the perioperative context is cardiac arrest (CA), which, according to studies, occurs more frequently during the induction of anesthesia than during the surgical procedure itself [9]. Although CA are rare overall, they have serious consequences, particularly for otorhinolaryngology patients and obese patients [9]. Neonates, infants, and children are a special group of patients who are particularly vulnerable to hypoxemia in the context of advanced airway management [10,11,12]. The incidence of intraoperative hypoxemia in neonates and infants is high and correlates with age. Documentation indicated an incidence of over 10% in older children, while rates of over 50% were observed in neonates [13]. The aim of this study was to establish a prospective comprehensive airway registry that enables the broadest possible recording of advanced airway management in a heterogeneous patient population.

## 2. Materials and Methods

### 2.1. Study Design and Study Setting

This study is a prospective and single-center observational study, which took place at the University of Heidelberg, Medical Faculty Heidelberg, Department of Anesthesiology, Germany. Our hospital is a university maximum care hospital with 2.599 beds. The Department of Anesthesiology provides comprehensive anesthesiologic services and oversees the intensive care units, offering a full range of intensive care therapies, including extracorporeal procedures. All specialist medical departments are represented within the university hospital. There was no randomization. It was a purely descriptive design, without intervention or influence on patient care.

A case report form (CRF) was attached to each anesthesia documentation during the study period. The CRF was completed by the anesthetists themselves immediately following the securing of the airway. In the event that the anesthetists’ documentation was incomplete, the patient was not entered in the register. Furthermore, it is possible that the anesthetists in attendance did not complete the CRF, which may result in a certain number of patients being excluded from the register. However, as the exact number of anesthesia’s performed in the specified period at our clinic was known to the accounting office, the number of patients who could not be included for the reasons mentioned above can be accurately determined. As there was no further follow-up, we did not anticipate that any restrictions will be encountered.

#### Ethics Approval

This study was conducted in accordance with the Declaration of Helsinki and approved by the Ethics Committee of University of Heidelberg, Medical Faculty Heidelberg, Germany (protocol code S-503/2023, date of approval 11 October 2023). As part of this study, the patient-related basic data (age, weight, etc.) were immediately anonymized by the treating physicians and recorded on the CRF. Accordingly, no personal data, such as date of birth or place of residence, were typically recorded. As a result of the immediate anonymization of the CRF by the physicians and the subsequent central collection of the CRF, it was not feasible to re-identify the individual patients. Consequently, in alignment with the Ethics Commission of the Medical Faculty of the University of Heidelberg, informed consent was not obtained from the patients.

### 2.2. Methodological Goalss

This study systematically documented and compared different methods of advanced airway management. The primary objective of the research project was quality assurance and quality development, and thus consecutively patient safety. By evaluating the in-house performance of medical staff, quality indicators were left to be identified later, as well as conclusions drawn about the need for training and supervision in individual vulnerable areas. This approach is already an established procedure in other countries for evaluating patient safety in the context of anesthesia and airway management [9].

### 2.3. Inclusion Criteria

All patients who required general anesthesia with consecutive advanced airway management (tracheal intubation and extraglottic airway management), as part of a surgical procedure at Heidelberg University Hospital in the first study period from 15 February 2024 to 31 March 2025, were included. Gender distribution could not be influenced.

### 2.4. Exclusion Criteria

No exclusion criteria.

### 2.5. Intervention Description

The general medical history data were collected in an anonymized case report form (CRF). Medical history was recorded once by the medical staff on a separate CRF, which contains 146 items, whereby complete anonymization was guaranteed. Detailed personal data such as name, date of birth, or address were not collected. The following information was collated in accordance with the CRF, as detailed below:

#### 2.5.1. Information on Medical Staff

Gender;Current year of further training;Years as a specialist physician;Years as a senior physician;Additional qualification in intensive care;Additional qualification in emergency medicine.

#### 2.5.2. Patient Characteristics, Diseases, and Complications

The relevant items of the CRF are documented in detail by the physicians, as demonstrated in Table 1.

### 2.6. Medical Equipment

For all tracheal intubations, we used laryngoscopes from HEINE Optotechnik, Gilching/Germany. Both Macintosh and Miller blades are available for conventional laryngoscopy. Stylets were used from AEROtube^®^/Lünen, Germany. All video-laryngoscopic intubations were performed using the C-MAC^®^ system (Karl Storz, Tuttlingen, Germany). Both Macintosh-style and hyperangulated blades (DBLADE^®^) were available. A variety of Dräger respirators (Lübeck, Germany) were used to ventilate patients.

### 2.7. Criteria for Discontinuing or Modifying Allocated Interventions

Study participants (physicians) could discontinue this study at any time without giving reasons, for example, if a physician terminated their employment or moved to another hospital. As a university hospital, our hospital attracts employees with a keen interest in scientific work. This motivates participants to take part in the study. A member of the study team was always available to answer any questions that participating physicians may have.

### 2.8. Relevant Concomitant Care Permitted or Prohibited During the Trial

No concomitant treatments were excluded or prohibited as part of this study. This also applied to surgical procedures, medical treatments, or other forms of therapy administered to the patient. However, this study had no influence on these concomitant treatments. Of course, this information was documented (e.g., type of operation, etc.).

### 2.9. Liability Insurance

The project was covered by the business liability insurance of Heidelberg University, Faculty of Medicine Heidelberg, Germany.

### 2.10. Sample Size and Data Management

Due to the study design and the establishment of a continuous airway registry, it was not possible to plan the number of cases. Instead, the exact number of cases will be planned within the framework of each individual project from the registry and specified in the respective publications. Due to organizational aspects, we plan to include approximately 9000 patients in the first phase of the registry.

#### 2.10.1. Data Management

The study stages were documented in compliance with German and European data protection laws. All data identifying the patient were only accessible to the attending medical and nursing staff as part of routine care and were not recorded in this study. It was, therefore, not possible to re-identify patients or physicians. For legal reasons, the anonymized study data (study questionnaire) will, nevertheless, be stored in the archives of the Department of Surgery for a period of 10 years.

#### 2.10.2. Confidentiality

The sensitive data (informed consent form) of all physicians were archived, locked, and inaccessible to others. All anonymized original documents will be kept locked in the clinical research unit of the Department of Anesthesiology for the next 10 years after publication. The study data will be handled in accordance with the German Federal Data Protection Act, which implements Directive 95/46/EC on data protection (Data Protection Directive). The electronic study database is anonymized and will also be stored for 10 years after publication.

#### 2.10.3. Statistical Methods

A statistical analysis was then carried out using SPSS Version 29.0 (Statistical Product and Services Solutions, SPSS Inc., Chicago, IL, USA). Detailed descriptive statistics of all collected data were performed. The data were checked for normal distribution using the Kolmogorov–Smirnov test. The univariate statistics were carried out using either the T-test (normally distributed data) or the Mann–Whitney test (non-normally distributed data) for continuous variables and scores. The chi-square test was used for the evaluation of categorical variables. A logistic regression model was calculated, and an ROC analysis with the calculation of cut-off values was carried out to check the prognostic significance of the individual markers.

### 2.11. Adverse Event Reporting and Harms

The potential for adverse events exists independently of our study of every medical procedure. As this is an observational study, there is no study-related intervention, and thus no harm is to be expected from this study.

### 2.12. Trial Status

Protocol version: 1.0 November 2024. Ethical approval: 11 October 2023 (Project-ID: S-503/2023) Recruitment initiation: 15 February 2024; Anticipated recruitment finalization: 31 March 2025.

## 3. Expected Results and Outcomes

### 3.1. Outcomes

#### Outcome Details

Creating an airway register that is as diverse as possible with a heterogeneous patient population and a wide variety of airway management methods. The real-world data generated in airway management can be used to reliably record incidences, methods, and complications and possibly derive recommendations for airway management. Table 2 shows the definition of the individual outcome parameters.

### 3.2. Objectives

“First-pass-success” and “overall-pass-success” of residents;“First-pass-success” and “overall-pass-success” for specialists and consultants;Difference in first-pass-success in different patient groups;Incidence of anesthesia-related complications with regard to airway management (e.g., aspiration, hypoxia, circulatory failure);Chosen exit strategy in case of difficulties;Incidence of difficult mask-bag ventilation;“First-pass-success” and “overall-pass-success” of extraglottic airway devices;Failure rate of extraglottic airway devices.

### 3.3. Participant Timeline

This study will approach all anesthesiologists of the University Heidelberg and offer them the opportunity to participate. The first study period will continue until the 31 March 2025, with an estimated number of around 9000 patient cases. It is intended that a specific period be documented on an annual basis, with the registry undergoing continuous updates.

### 3.4. Dissemination Plans

The findings will be shared with the scientific community and relevant groups through publication in scientific peer-reviewed journals, conference presentations, and reporting on databases such as ClinicalTrials.gov and social media.

## 4. Discussion

To the best of our knowledge, the HAPA-registry is the first German registry to collect routine data on advanced airway management of all patient groups, including pediatrics. Endotracheal intubation (ETI) represents a fundamental skill for anesthetists and other physicians [14]. The procedure is employed in a variety of settings, including during surgical procedures, in the intensive care unit, during periprocedural anesthesia, and in emergency medicine. The clinical relevance of airway management is particularly evident in the fact that the primary cause of significant anesthetic-related complications can be attributed to issues pertaining to airway management [9]. Despite the increasing technological development in recent years (VL, video endoscopy, tracheal tubes with integrated camera), the clinical relevance of the individual methods and procedures was not conclusively scientifically substantiated. In order to enhance patient safety, it is imperative that comprehensive documentation and structured training in airway management and the assessment of outcome variables are implemented. The aim of this study is to collect “real-world data” on advanced airway management and its complications in different patient cohorts over a large period at the University of Heidelberg, Medical Faculty Heidelberg, Department of Anesthesiology, Germany. This is designed to serve as an internal quality control mechanism within the hospital, with the objective of ensuring the delivery of high-quality care. Anonymized data may be employed to investigate the incidence of complications, the success rate of the initial intubation attempt, otherwise known as the ‘first-pass success’ rate, and other clinically relevant parameters. The data generated could be employed to develop recommendations for instructions in elective airway management and to initiate optimizations for patient safety as part of our own quality assurance process. Various studies in the past showed that the successful completion of intubation on the first attempt is of crucial importance in minimizing complications and time losses [15,16]. First-pass success is influenced by various factors, particularly the experience of the medical staff and the choice of airway management method used [17]. Studies showed that a higher number of intubation attempts correlates negatively with patient outcomes. In addition, first-pass success rates and complication rates vary significantly between different medical specialties [9,18]. The establishment of the prospective HAPA-registry could facilitate the analysis of advanced airway management outcomes across different medical specialties. It would be beneficial to identify common complications and to investigate differences in first-pass success depending on the specialty and the experience of the staff. Furthermore, it would be favorable to analyze which tools are frequently employed in specific patient groups and whether there are particular methods that are more or less effective in certain populations.

## Figures and Tables

**Table 1 mps-08-00006-t001:** The table shows a selection of items from the case report form (CRF).

Patient Characteristics	Diseases	Complications
Sex, body weight, body height	Coronary heart disease	Tooth damage
ASA physical status	Peripheral arterial disease	Oro-/pharyngeal bleeding
Mallampati score	COPD/Asthma	Regurgitation
Limited mouth opening, <3 cm	Diabetes	Aspiration
Obstructive sleep apnea	Hypertension	Foreign body obstruction
Reduced cervical mobility	Dyslipidemia	Laryngospasm
Cormack/Lehane score	Nicotine abuse	Desaturation/hypoxia
Percentage of glottic opening (POGO)	Alcohol abuse	Hypotension
	Kidney failure Cancer	

**Table 2 mps-08-00006-t002:** Definition of the important outcome parameters.

First-Pass Success	Desaturation	Hypotension
The initial tracheal intubation attempt was defined by the insertion of a laryngoscope blade and/or tracheal tube into the patient’s mouth The initial intubation attempt was considered to have failed if it did not result in successful endotracheal intubation, with or without an attempt to pass the tube Subsequent attempts at intubation were characterized by the reinsertion of an endotracheal tube or the insertion of the same or a new laryngoscope blade	<90% O_2_ saturation if not pre-existing and without a prescribed time period A single valid measurement <90% is documented with the term “desaturation”	<65 mmHg means arterial pressure if not pre-existing and without a prescribed time period

## Data Availability

The data that support the findings of this study are available on request from the corresponding author, D.D.U.

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
