# Peer review of "Heidelberg Adult and Pediatric Airway Registry (HAPA-Registry)"

_mps, 2025, doi:10.3390/mps8010006_

Round 1

Reviewer 1 Report

Comments and Suggestions for Authors

1. The introduction section is lengthy and could benefit from being more concise. Consider summarizing key points to improve readability and focus on the study's unique contributions.

2. The inclusion criteria state:
“All patients who require general anesthesia with consecutive advanced airway management (tracheal intubation and extraglottic airway management) as part of a surgical procedure at Heidelberg University Hospital in the study period (15.02.2024 - 31.03.2024) are included.”

  • Is the study period restricted to 15.02.2024 - 31.03.2024? This point requires clarification, as it conflicts with later descriptions of a longer recruitment timeline.
  • Additionally, correct the grammatical error: replace "an surgical procedure" with "a surgical procedure" (line 105).

3. The patient information collected appears minimal. Consider expanding the dataset to include variables if possible, such as:

  • Presence of facial hair, dentures, or missing teeth
  • Restricted head and neck mobility
  • History of obstructive sleep apnea (OSA)
  • Micrognathia or limited mouth opening

4. In section 2.10, the trial status mentions:
"Protocol version: 1.0 November, 2024. Ethical approval: October 11, 2023 (Project-ID: S503/2023). Recruitment initiation: February 15, 2024; Anticipated recruitment finalization: March 31, 2025." This timeline appears inconsistent with the study period (15.02.2024 - 31.03.2024) mentioned in the inclusion criteria. Clarification is needed to reconcile these discrepancies.

5. Outcomes Section

  • If the outcomes are not divided into primary and secondary, consider consolidating them under a single "Outcomes" heading for clarity.
  • Outcomes should ideally be listed as statements (e.g., "Incidence of complications") rather than questions.
  • Additionally, provide precise definitions, particularly for terms such as "first-pass success", to ensure consistency in data interpretation.

6. Clarify how and when informed consent is obtained from patients. This detail is essential for demonstrating ethical compliance.

7. The ethics section requires further elaboration on how patient anonymity is maintained while handling the 146 data points in the CRFs. While anonymization is mentioned, providing a specific description of techniques such as encryption, data masking, or controlled access would reassure readers about compliance with privacy standards.

8. The manuscript does not explain how cases with incomplete data or patients withdrawing during follow-up will be managed.The authors should outline a clear strategy for addressing these scenarios, including their impact on data integrity and how biases will be minimized.

Author Response

Dear Reviewer please see the attachment. Thank you for your work. 

Reviewer 2 Report

Comments and Suggestions for Authors

Thank you for giving me the opportunity to review this article called: “Heidelberg Adult and Pediatric Airway Registry (HAPA-Regis- 2 try)” it is the protocol in order to achieved a monocentric prospective register to assess the compilcations related to airway management.

The project is well written and subject is understandable.

I have few remarks:

Can you shorten the introduction

Do you planned to collect the data via a eCRF.

How you will be sure the include all the patients?

Do you have already those information in your computerized medical reports?

The duration and period of inclusion must be the same into the manuscript. Do you plane to open this registery over 1 month, or 1 year, when do you plan to start the inclusions?  

Could you be more precise for the risk factors of intubation difficulty (used of the items of the MACOCHA score?).

Please collect the details of the material used for intubation, give the details of the intubation failure and of the complications related to the intubation.

Author Response

Please see the attachment. Thank you very much for your work.

Round 2

Reviewer 1 Report

Comments and Suggestions for Authors

5. Outcomes Section

If the outcomes are not divided into primary and secondary, consider consolidating them under a single "Outcomes" heading for clarity.

Outcomes should ideally be listed as statements (e.g., "Incidence of complications") rather than questions. Additionally, provide precise definitions, particularly for terms such as "first-pass success", to ensure consistency in data interpretation.

Thank you for this important information. We have changed the heading to Outcomes!

Furthermore, we have defined the important outcomes and presented them transparently in our table. We hope that this will make it easier to understand.

Reviewer reply:

I believe the authors’ definitions of important outcomes are still simplistic and require more detailed descriptions. Additionally, the authors’ definition of “First-Pass Success” as “Laryngoscope blade passes the lips” seems more like a description of the “first attempt at insertion.” Please carefully recheck the accuracy of this definition.

Similarly, the definition of desaturation also needs clarification. Does it refer to an O2 saturation drop below 90% for more than 3 seconds or 5 seconds, or does it simply mean any value below 90%?

Furthermore, regarding the definitions of outcomes and the selection of result variables, the authors may find this article helpful as a reference: (JAMA. doi:10.1001/jama.2024.0762).

Author Response

Dear Reviewer, 

Tank you for your efforts regarding our re-submission. For our point by point pleas see the attachment. 
